# An integrated self-optimizing programmable chemical synthesis and reaction engine

Artem I. Leonov[1,2], Alexander J. S. Hammer [1,2], Slawomir Lach [1], S. Hessam M. Mehr [1], Dario Caramelli[1], Davide Angelone [1], Aamir Khan[1], Steven O'Sullivan[1], Matthew Craven[1], Liam Wilbraham[1] & Leroy Cronin [1] ✉

Robotic platforms for chemistry are developing rapidly but most systems are not currently able to adapt to changing circumstances in real-time. We present a dynamically programmable system capable of making, optimizing, and discovering new molecules which utilizes seven sensors that continuously monitor the reaction. By developing a dynamic programming language, we demonstrate the 10-fold scale-up of a highly exothermic oxidation reaction, end point detection, as well as detecting critical hardware failures. We also show how the use of in-line spectroscopy such as HPLC, Raman, and NMR can be used for closed-loop optimization of reactions, exemplified using Van Leusen oxazole synthesis, a four-component Ugi condensation and manganese-catalysed epoxidation reactions, as well as two previously unreported reactions, discovered from a selected chemical space, providing up to 50% yield improvement over 25–50 iterations. Finally, we demonstrate an experimental pipeline to explore a trifluoromethylations reaction space, that discovers new molecules.

Smart laboratory automation holds promise to accelerate chemical research, eliminate tedious tasks, improve safety and reliability[1–4]. Recently there has been significant progress towards more automated synthesis platforms[5,6]: systems that can perform a large variety of synthetic processes[7–12], giving access to a diverse set of target compounds[7,9,13]. While these platforms perform elaborate experiments in a fully automated fashion, they are limited to sequential processes, adapted from literature and trivial laboratory operations. The lack of real-time data and feedback control does not allow for self-correction and dynamic process execution. Acidifying a reaction mixture to a certain pH or maintaining the internal reaction temperature during oxidant addition are trivial for a human researcher but challenging and crucial for the safe operation of automated laboratory equipment. While condition monitoring and process control are routine tasks in the chemical and pharmaceutical industry[14,15], it is much less common in academic research laboratories, with much data reliant on human intervention (e.g.

visual inspection) not captured. The ability to intelligently select and perform experiments, however, is key to fully leverage the potential of robotic systems in the chemical domain[16].

Reaction optimization, benefitting from incorporation of the analytical data into the workflow, has become a part of the chemical automation development[17–21]. However, the vast majority of published platforms are limited to narrow chemical tasks with few exceptions of systems for flow chemistry[22–24]. These systems, while demonstrating proof-of-concept results, are bound to specific hardware modules and software, thus making the optimal protocols not transferable across chemical automation robots. In this work, we build upon our universal abstraction of chemical synthesis, chemputation, by describing how process sensors and analytical instruments can be coupled with our chemical processing unit (Chemputer)[25]. This allows for the autonomous execution and optimization of literature protocols. Telemetry data is used for process state monitoring and with predefined rules allows for dynamic procedure execution, self-correction and real-time

[1]School of Chemistry, The University of Glasgow, University Avenue, Glasgow G12 8QQ, UK. [2]These authors contributed equally: Artem I. Leonov, Alexander J. S. Hammer. ✉ e-mail: lee.cronin@glasgow.ac.uk

decision making. We show how the system can react to the changing environment in an adaptive temperature-controlled thioether oxidation, a colour-monitored nitrile formation and in case of a critical liquid handling platform failure. Furthermore, when coupled with analytical instruments, capable of quantifying reaction outcomes, dynamic execution is used to create a closed-loop system for reaction optimization, see Fig. 1.

This framework is built on top of the existing abstraction of chemputation, which is encoded using the χDL programming language[26], thus enabling iterative optimization on any hardware platform capable of performing the relevant chemical unit operations (e.g. reagent addition, stirring with temperature control, etc.). We demonstrate the system's usability for reaction optimization by improving the product yield and purity for the 4-component Ugi reaction; Van Leusen oxazole synthesis; manganese-catalysed styrene epoxidation and explorative trifluoromethylation using the Ruppert–Prakash reagent. By using a unified format for storing and sharing procedures, process data and results we ensure that every protocol can be reproduced and verified. The key requirement for any autonomous chemical robot is the ability to dynamically execute a given list of instructions with real-time adaptation to changing process parameters. To realize this on the Chemputer platform, the following components were integrated in the overall framework: (a) hardware and software support for a range of low-cost sensors, (b) dynamic χDL as a basis for various feedback control chemical operations, (c) software package for analytical instrument control and signal processing, (d) χDL-based package for iterative reaction optimization with support for parallel procedure execution. These improvements enabled the first demonstration of an automated tandem of discovery-optimization framework that uses XDL code as an input and returns optimized XDL as an output, paving the way for fast collaborative exploration of chemical spaces and reaction conditions.

## Results and discussion

We have included a set of low-cost sensors into the existing infrastructure of the Chemputer platform: colour, temperature, conductivity, and pH sensors for monitoring of chemical processes, and a liquid sensor for tracking material transfer and detecting failures of the liquid handling system. An environmental sensor was added to record the ambient conditions—temperature, pressure and humidity—and identify potential reproducibility issues. All sensors are connected to a custom-designed board, the *SensorHub*: an Arduino module, featuring a variety of communication protocols and connected to the Chemputer IP network (ESI Section 3). Additionally, graphical interface is provided through a web-based dashboard application (Fig. S18), which allows the user to control any sensor individually, or change the rate of the background measurements for demanding processes. In addition to the low-cost sensors, a vision-based condition monitoring system was developed to add flexibility and improving the autonomy of Chemputer operations.

To manage the control of the analytical instruments and provide a unified interface for obtaining spectral data, we have developed a stand-alone Python package—*AnalyticalLabware*. Covering a range of several analytical instruments, our library includes control over UV-Vis, near IR (NIR), Raman and NMR spectrometers as well as HPLC-DAD system from various manufacturers, see Fig. 2. In addition, the library also has basic methods for spectra pre-processing, such as peak picking or baseline correction, and domain specific techniques, e.g. zero-filling and apodization for NMR spectra (ESI, Section 2). The package is fully integrated into our Chemputer workflow: from hardware graph, in which the instruments are presented as hardware objects with corresponding connection parameters, to dedicated high-level χDL steps, specifying sampling routine and additional parameters to perform the data acquisition.

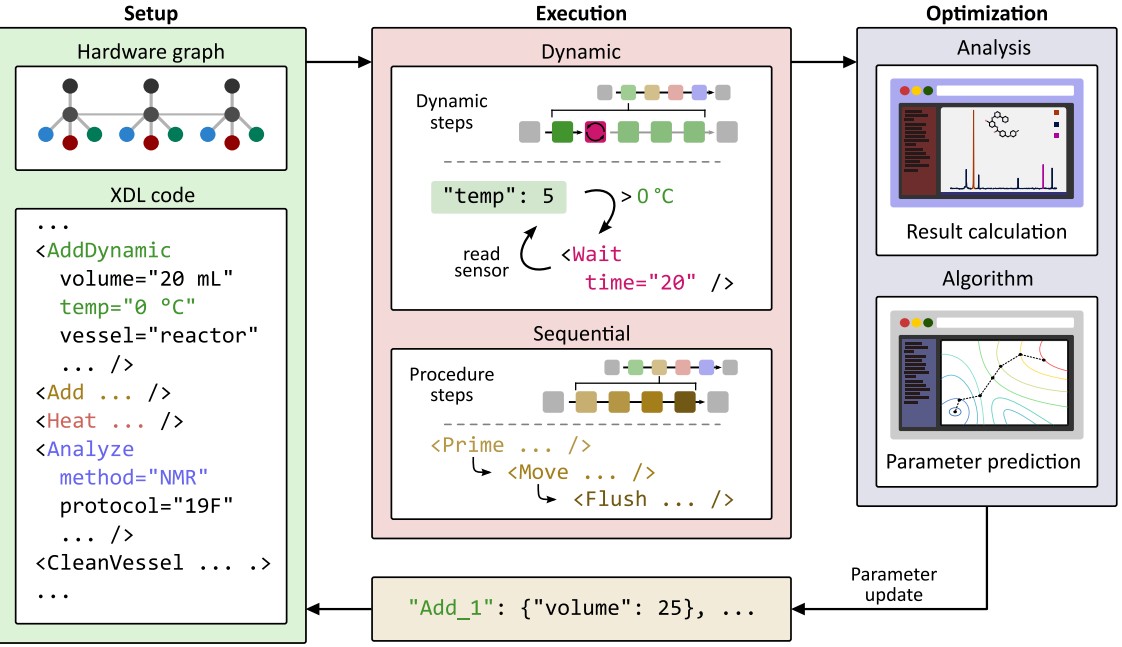

**Fig. 1 | Overview of the dynamic chemical operations execution.** The procedure, illustrated using χDL syntax, may be executed sequentially one step after another or dynamically, e.g. as highlighted with AddDynamic where a Wait step is inserted if the temperature rises above given threshold. With integrated analysis and corresponding χDL step (Analyze), the procedure could run iterative with dynamic update of its parameters, thus allowing for closed-loop reaction conditions optimization. The Setup step represents the definition of the hardware configuration of the system and the chemical procedure defined digitally in the χDL code. In the Execution step these components are used to operate the physical robotic system to achieve the defined chemical operations, including the collection of analytical data about the execution. In the Optimization step, the data is analysed and algorithms are used to predict parameters for the next round of experiments.

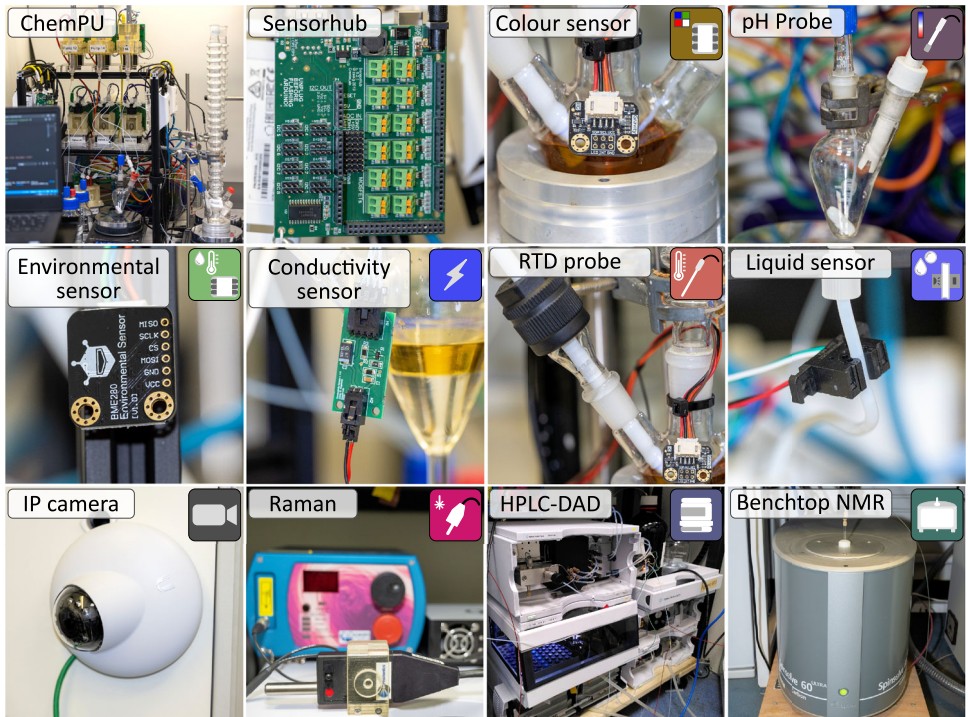

**Fig. 2 | The suite of sensors and analytical instruments integrated in the Chemputer stack.** The *SensorHub* provides a unified interface for a range of low-cost sensors. Sensors were used to monitor colour, pH, ambient conditions (pressure, humidity and temperature), internal temperature (Resistance Temperature Detector (RTD) probe), conductivity and liquid transfers. An Internet Protocol (IP) camera was integrated for video capture and active failure detection. Analytical instruments (Raman, HPLC-Diode Array Detection (DAD), NMR) were integrated for reaction monitoring and optimization.

At the fundamental level χDL provides a universal ontology for the encoding and execution of chemical synthesis recognizing that all chemical synthesis is based around four abstract properties: the reaction; workup; isolation; purification. This means that well known chemical reactions can be expressed as a process-driven language that focuses on the practical actions needed to allow the reaction to happen. To extend this to dynamic reaction control, a base class for dynamic processes, AbstractDynamicStep, is exposing three abstract methods to control the execution flow, where each method returns a list of steps to be executed, based on the current state of the step. Here, we present a set of dynamic χDL steps to allow self-correcting procedure execution for a range of potential use cases (addition, transfer, execution, monitoring, optimization).

The *ChemputationOptimizer* software is designed to take further advantage of the χDL dynamic step by leveraging a set of optimization algorithms to dynamically update the procedure parameters based on an end-point measurement obtained from a given analytical instrument (Fig. 3). The χDL procedure, either translated from a literature or obtained followed combinatorial or active learning reaction discovery, is used as a starting point for the optimization cycle. The user only needs to provide a corresponding hardware graph and a configuration file. After robotically executing the procedure, the reaction output (typically, a spectrum of the quenched reaction mixture) is analysed and passed to an optimization algorithm to suggest the next set of input conditions. The user can choose from a wide variety of state-of-the-art optimization algorithms, including those implemented in the Summit[27] and Olympus[28] frameworks. A server–client interaction allows multiple clients to work together towards a joint optimization problem. With the new set of reaction parameters, the procedure is updated and executed, and this cycle is repeated until the maximum number of iterations, or the desired target is reached. All experimental procedures together with the corresponding set of parameters and reaction results (both raw spectral data and the processed output) are saved in a database and can be verified later.

With these new hard- and software developments in place, we validated that low-cost sensors can capture relevant process data of the synthesis execution, ensuring safety and stability of operation. Our experience showed that the most encountered critical failure within the Chemputer is related to various types of syringe breakage. For such potential hardware failures, we employed the vision-based condition monitoring system that uses multi-scale template matching, detects anomalies using a holistic approach of structural similarity, and alerts the operator (ESI Sections 3.3.1 and 5.3.1).

We deployed our data-rich reaction development engine to passively monitor the turbidity during the formazine synthesis, an organic colloid used as a turbidity reference material (Fig. 4a, ESI Section 5.3.3). The system did not only capture ambient conditions, e.g. the temperature of the room. Also reagent priming and their addition left distinct signals in the liquid detector trace, while the increase of turbidity as the reaction progressed has been detected by the RGBC sensor. The liquid sensor was further used to monitor the consistency of reagent delivery and proved valuable in challenging steps such as filtration, where transferring volumes are not pre-defined. Here, a simple binary output (i.e. 0 for an empty tubing and 1 for filled) was sufficient to increase the reliability of the process, while quantitative volume data might be obtained via indirect analysis (ESI Section 5.3.2). Overall, combined data could serve as a process fingerprint and may be used for subsequent validation of any reproduced procedure (ESI Section 5.3.4).We have demonstrated the self-correcting execution of two functional group interconversion reactions (thioether oxidation[29] and nitrile formation[30]), using the feedback from the temperature and colour sensors, respectively. The examples were chosen to highlight the benefits of incorporating the selected low-cost sensors in automated synthesis platforms. The first example shows the slow addition of hydrogen peroxide monitored by an internal temperature probe to prevent the thermal runaway. By utilizing the dynamic step, it was possible to carry out the reaction automatically on a 25-g scale without exceeding the maximum

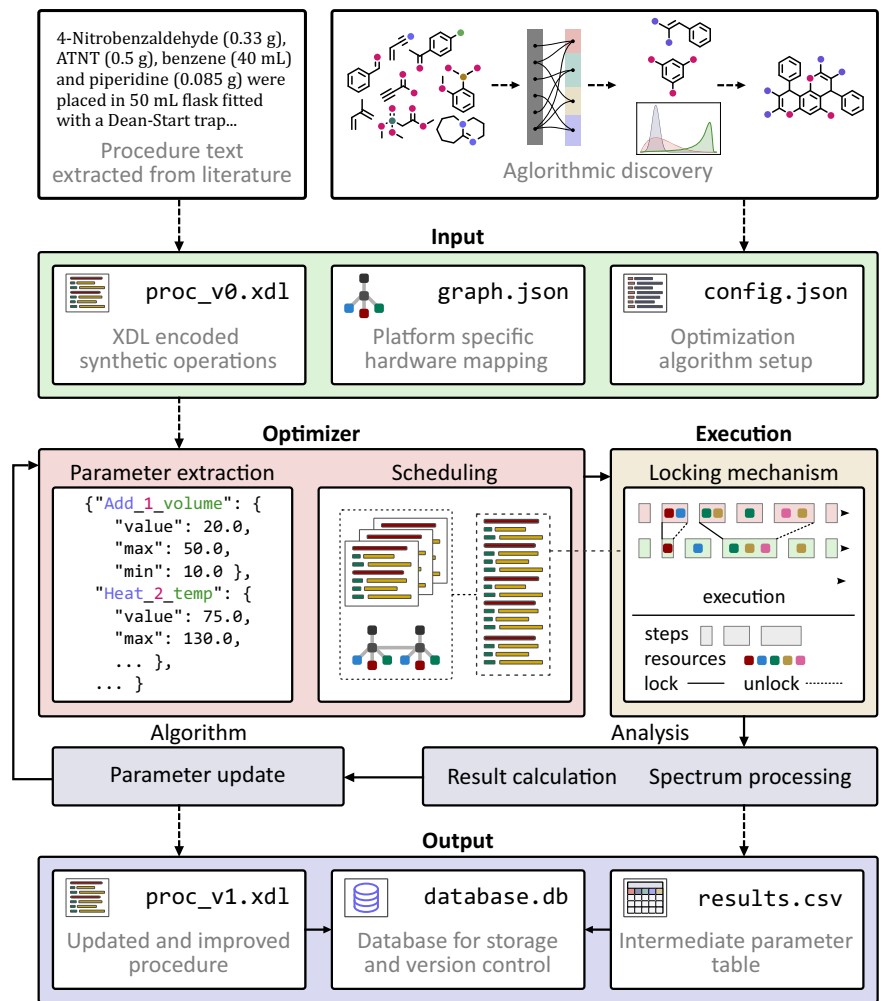

**Fig. 3 | An overview of the framework for chemical discovery and optimization, *ChemputationOptimizer*, its system architecture and operation.** The initial χDL procedure (proc_v0.xdl) is obtained either via text translation from literature or upon algorithmic reaction discovery. Together with the hardware graph (graph.json) and a configuration file (config.json) are loaded, from which the optimizer framework extracts the parameters to be optimized. If enough resources are available, the optimizer performs the scheduling routine to allocate hardware resources to the corresponding procedure steps for parallel execution, minimizing the total duration. A locking mechanism which ensures error-free execution during runtime and eliminates the risk of unexpected cross-contamination. Upon successful execution, the outcome of the reaction is analysed quantitatively and the results are fed to the optimization algorithm to suggest the next parameter set and update the initial procedure. The updated procedure, together with the results table is saved, using the database. The architecture and implementation details are given in the ESI (Section 1).

temperature specified in the literature procedure during the oxidant addition step (Fig. 4b).

Next, the passive temperature monitoring revealed an uncontrolled exotherm during the subsequent heating step. Such insights can be easily discerned through our dashboard (ESI Section 3.4), enabling safe process development and scale-up. Our second example demonstrates the use of a simple colour sensor to monitor a nitrile synthesis from an aldehyde using ammonia and iodine, and dynamically adjust the reaction time as the discolouration indicates complete reagent consumption (Fig. 4c).

The reaction time of the chosen process varies depending on the aldehyde substrate, which must be determined using a supervised trial run if in-line feedback is not available. Typically, exact reaction times are unknown and require a more general solution, such as the one outlined in our χDL step for dynamic execution which performs the respective child step(s) until termination criteria are met. In our experiments, absolute thresholds for sensor readings proved unreliable as termination criteria due to changes in the ambient light, however we found that rate of colour change could be used to detect the end of the process. Beyond enhanced process control and condition

monitoring, we were able to optimize reaction conditions for multicomponent[31], heterocycle synthesis[32], and catalytic reactions[33] using feedback from [19]F NMR, HPLC-DAD, and Raman spectroscopy. The goal was to illustrate a broad change of chemistry that features commonly encountered reaction types in preparative synthesis to showcase the benefits of digitalization for the traditional organic chemistry community. The algorithms were chosen to highlight the capabilities of our agnostic toolkit (integrated optimization routines using random search, design of experiments, Bayesian optimization and genetic algorithms as well as interface to the Summit and Olympus frameworks) and a user may choose the most appropriate algorithm for their chemical system. In our experience, a wide variety of algorithms were suitable for reaction optimization and even random search provides a strong baseline. As a proof-of-concept for reaction optimization, we have selected the Ugi four-component reaction (Fig. 5a). Even though the reaction procedure is well established, it is sometimes not trivial to obtain good results, given that multiple parameters need to be optimized simultaneously. The target parameter for this reaction was selected as the ratio between the area under the curve for the product and reference substance on the [19]F

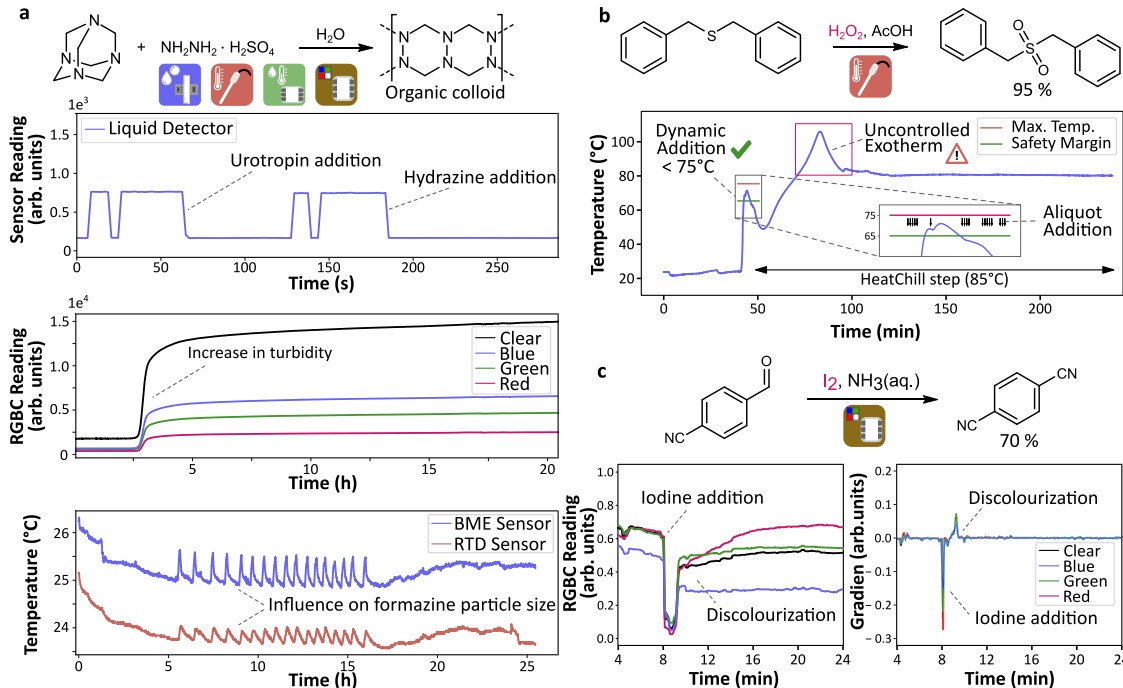

**Fig. 4 | Reaction monitoring in the automated synthesis execution. a** Passively monitored formazine synthesis, a turbidity reference material, **b** dynamically executed thioether oxidation and **c** iodine-mediated nitrile formation with corresponding exemplary data captured within the process. **a** Example data was captured via passive monitoring during the formazine synthesis: The liquid detector tracked reagent addition, the colour sensor (Red Green Blue Colour (RGBC)) could detect the increase in turbidity as the colloidal suspension formed, and the environmental (from a BME280 combined humidity, pressure and temperature sensor−BME) and internal reaction temperature (RTD) sensors could capture data relevant for reproducibility. **b** Internal reaction temperature was

captured in real time and used to control the addition of the oxidizing agent in the highly exothermic thioether oxidation reaction. The original procedure demanded to keep the internal reaction temperature below 75 °C. After the addition, passive monitoring further revealed an uncontrolled exotherm upon heating the reaction mixture to the temperature (85 °C) specified in the literature procedure. **c** A colour sensor captured data during the addition of iodine and its subsequent consumption as the reaction proceeds. The gradient was calculated post hoc, clearly showing the iodine addition and consumption as peaks. Source data are provided as a Source data file.

NMR spectrum. In 30 experiments, it was possible to achieve a 38% relative improvement for the yield of the product, compared to the starting literature conditions.

The optimization was performed using a Sequential Model-Based Optimization algorithm (SMBO) and consisted of four stages: five random experiments to initialize the parameter search space, 14 experiments to explore this space (i.e. minimize the uncertainty), five experiments with balanced exploration-exploitation approach and six experiments to exploit the space (i.e. maximize the outcome, product yield). A Gaussian Process regression model served as the surrogate model. Figure 5a shows a typical $^{19}$F NMR spectrum of the reaction mixture (bottom right) and explored reaction parameters space over the optimization process (bottom left).

Showcasing the flexibility of our approach, a Van Leusen oxazole synthesis was optimized using two parallel reactors with independent heating and stirring (Fig. 5b). The throughput of the platform can be increased by adding further reactor modules, however there is a trade-off with the number and the quality of updates obtained from the optimization algorithm. We used the SNOBFIT[34] algorithm to maximize the area of the product peak in the HPLC chromatogram relative to an internal standard while minimizing impurities and excess reagents achieving a 10% relative increase in the weighted objective in 26 iterations.

Finally, we optimized starting material conversion relative to an internal standard in a manganese-catalysed epoxidation (Fig. 5c) using online Raman spectroscopy in conjunction with the Phoenics[35] algorithm. Thanks to the fast acquisition time of the Raman instrument, time-series data could be obtained as well, which could be used for further analysis of the reaction kinetics. Not only classic process variables like temperature and time were considered but also often

overlooked factors such as addition speed proved crucial in this transformation as they impact the formation of the active catalytic species. Forty iterations were needed to find a robust optimum that leads to full conversion. We also demonstrated how this closed-loop approach can be extended to facilitate compound discovery and optimization. Our suggested pipeline includes three stages: first, the exploration of the product space achieved by a series of experiments from random search to algorithmic maximization of the heuristic novelty score (see ESI, Section 1.8). Next, the obtained spectra are analysed to isolate regions of interest and identify potential products. Finally, for each identified product a series of experiments is executed to find reaction conditions, that will maximize the product outcome. As an example of such exploration-optimization strategy, we have selected the trifluoromethylation reaction[36] within a small substrate space of four different starting materials (Fig. 5d). Starting from a complex mixture of substrates our system was able to identify three products of this reaction, which were characterized by corresponding peaks on the $^{19}$F NMR. These peaks were then used to guide the individual optimization experiments, where the target was set to maximize the area under the curve for the main product, while minimizing the areas for the other recognized peaks (full description is given in the ESI, Section 5.5.2).

Using combinatorial search in limited chemical space we were able to discover a reaction between toluenesulphonylmethyl isocyanide and benzylidenemalononitrile, referred later as the tosMIC reaction (Fig. 6a, see details in the ESI, Section 9). The χDL procedure generated upon the discover was subjected for the optimization using HPLC analysis feedback, set to maximize the product peak (ESI, Section 5.5.3). Using the SMBO strategy we were able to increase the yield

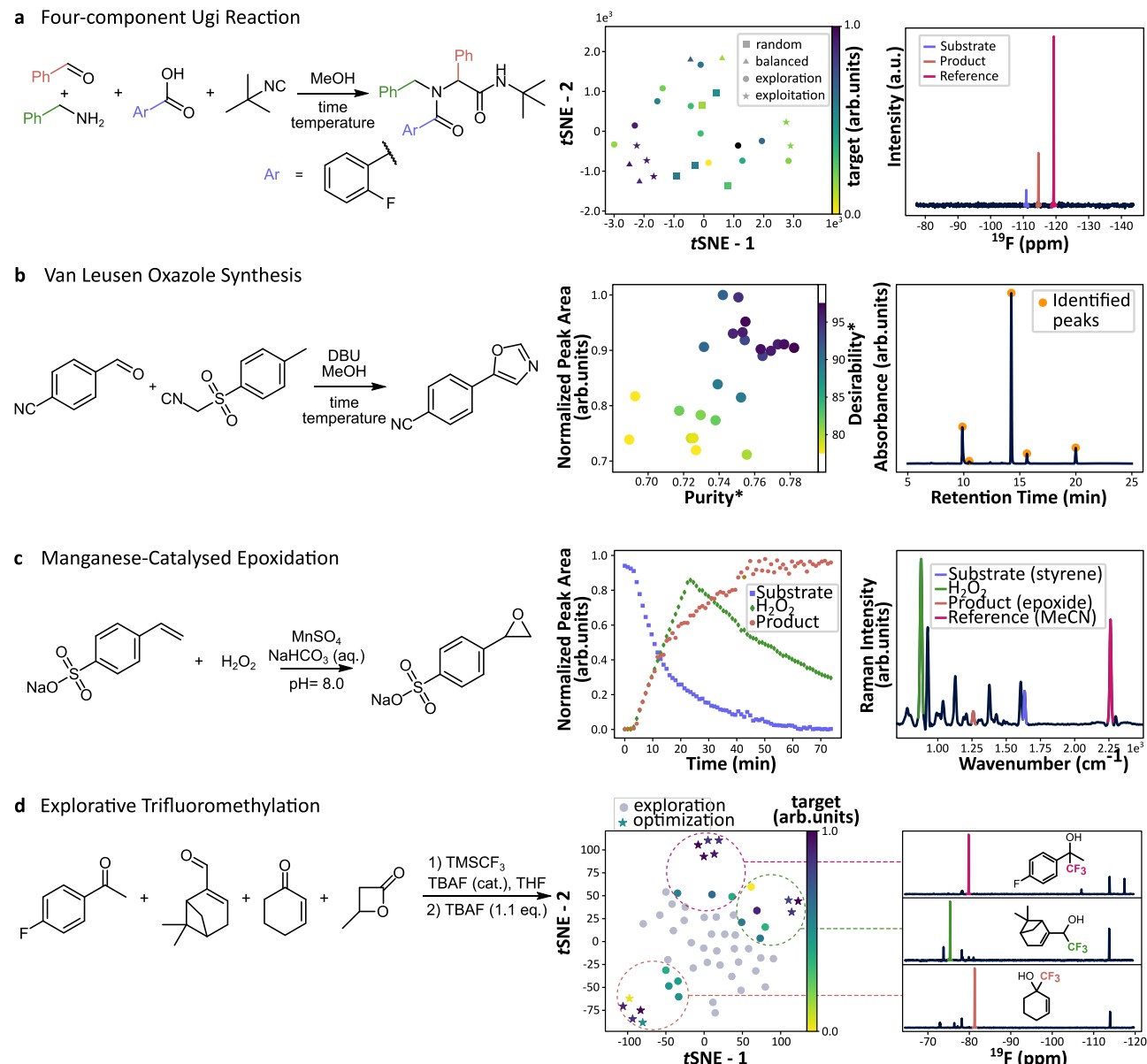

**Fig. 5 | Results of the closed-loop reaction optimization. a** Four-component Ugi reaction scheme (top); plot of the parameter space reduced to two dimensions using t-distributed stochastic neighbour embedding algorithm, where colour specifies the target parameter and the shape corresponds to the strategy used for the specific objective (bottom left); example of the $^{19}$F NMR spectrum of the reaction mixture (bottom right). **b** Van Leusen Oxazole synthesis scheme (top); optimization results plot, *: *purity* is the relative area of the product peak divided by the sum of all peak areas, *desirability* is calculated as the weighted average between the two objectives (see Supplementary Information, Section 5.5.2); example of the HPLC chromatogram (bottom right). **c** Manganese-catalysed epoxidation scheme (top);

example plot of the reaction monitoring using Raman spectroscopy (bottom left); example Raman spectrum (bottom right). Full description of the optimization parameters and chosen targets is given in the ESI (Section 5.5). **d** Trifluoromethylation reaction scheme; plot of the parameter space reduced to two dimensions using *t*-distributed stochastic neighbour embedding algorithm, where colour specifies the target parameter and the shape corresponds to the strategy used for the specific objective (bottom left); examples of the $^{19}$F Nuclear Magnetic Resonance (NMR) spectra where highlighted peak corresponds to the CF$_3$ group in the specified product (bottom right). Source data are provided as a Source data file.

by 22% in just 32 iterations (Fig. 6e). Furthermore, we decided to perform another reaction in a similar manner, this time, between three components: phloroglucinol, benzylidenemalononitrile and 1,8-bis(dimethylamino)naphthalene. The optimization protocol consisted of two distinct campaigns, the first one comprising 37 reactions and the second campaign, comprising additional 13 reactions in the expanded search space. The reason for such strategy originated from careful analysis of the data obtained for the first campaign: 6 out of the 37 reactions performed at 25 °C with maximally allowed 1,8-bis(dimethylamino)naphthalene concentration resulted in yields greater than the yield at initial reaction conditions (27%), suggesting that a

more "adventurous" approach may lead to interesting results[37]. The second campaign has been initiated with results from previous campaign serving as input and the temperature and constraints allowing for temperatures as low as 0 °C while allowing the 1,8-bis(dimethylamino)naphthalene concentration to reach 8 mol. During the first campaign the optimization protocol yielded parameters which resulted in a 49% yield, an absolute increase of 22% from the 27% obtained for the initial reaction conditions.

However, with the more opened constraints present, the explored space has been expanded into unintuitive areas that would not have been a first choice, or a choice at all, for the experimental chemist: in

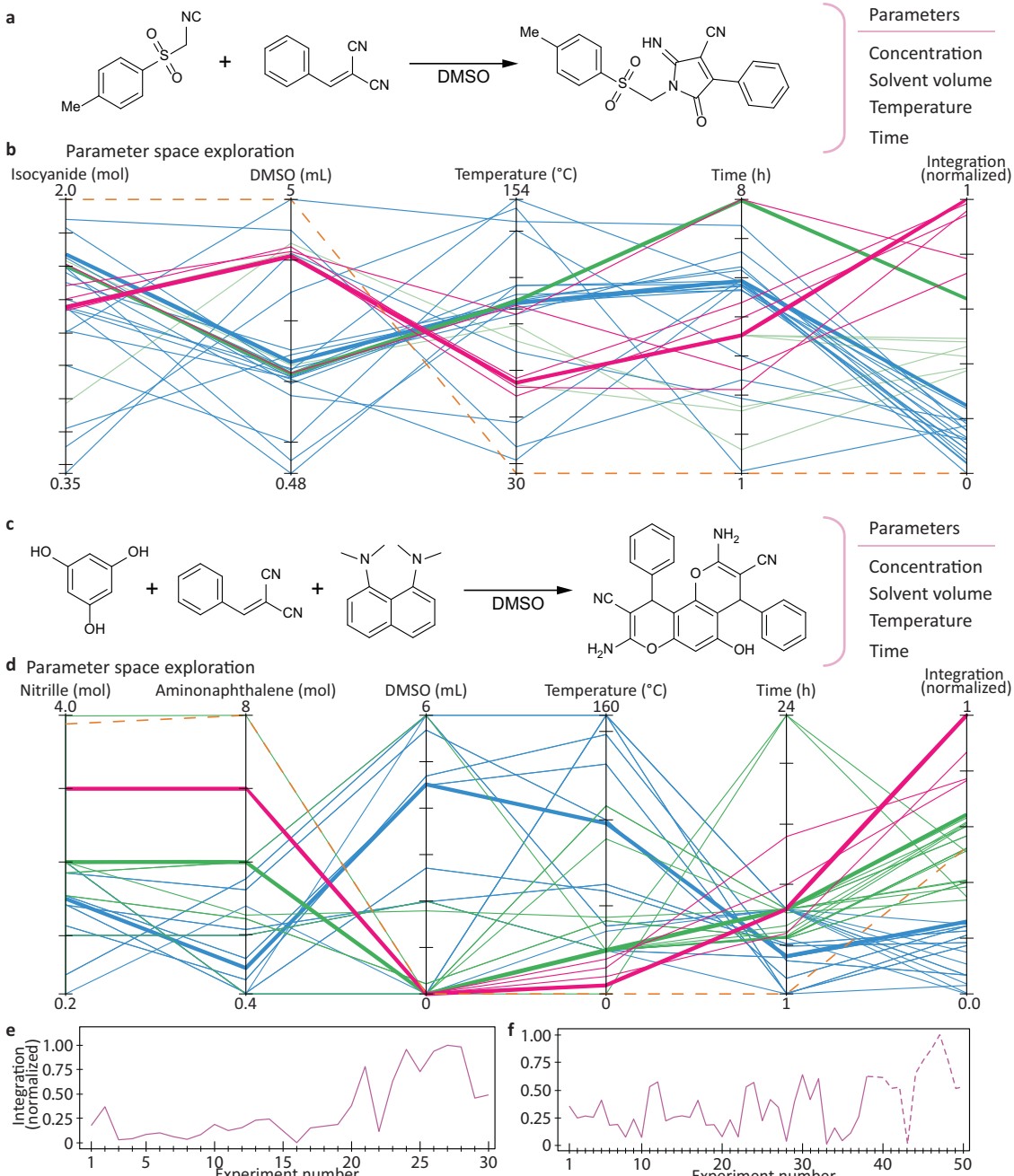

**Fig. 6 | Optimization examples.** General reaction schemes for the **a** tosMIC and **c** phloroglucinol reactions with denoted parameters being optimized during the optimization cycle; corresponding parallel coordinate plots (**b**) and (**d**) where each vertical axis represent one of above mentioned parameters altered during each reaction with the last axis denoting the amount of product detected. Starting from the left parameters are connected with a line which represents a single reaction run. The reactions have been divided into three arbitrary groups, coloured in red, green and blue, for ease of interpretation and visualization, with respect to the normalized integration obtained for each reaction (amount of product). The coloured bold line within each group denotes a reaction of highest normalized integration; the orange dashed line denotes initial reaction parameters applied for the first reaction. Change of the integration observed during the optimization process for the **e** tosMIC and **f** phloroglucinol reaction with the second campaign denoted by a dashed line. Source data are provided as a Source data file.

this example the algorithm moved into temperatures not suited for the solvent used (DMSO freezing point is below 19 °C). This shift has led to a further increase in the reaction yield by 50% with respect to the initial reaction conditions, reaching 77%.

In conclusion, low-cost sensors as well as process analytical technology instruments were integrated into the growing Chemputer software stack leading to enhanced process control and insight. This represents a fundamental shift from previous iterations of the Chemputer platform that were open-loop control systems for chemical

synthesis to a closed-loop platform with feedback-enabled synthesis, optimization, and discovery capabilities. In practice, this means that new dynamic operations such as temperature-controlled reagent additions or optical endpoint detection that were previously not available can now be used in automated synthesis, increasing safety and reliability of our system. Furthermore, we showed how χDL procedures derived from the literature can be improved and versioned using optimization and databasing. Executing the optimized, versioned XDL codes on our platform leads to reproducible optimal

results. Data thus obtained can be used to understand the influence of process variables on given chemical transformations which in turn may help to populate reaction blueprints with tailored conditions for library synthesis or accelerated molecular discovery projects. In this context, the newly developed χDL-based parallel execution and intelligent resource allocation have the potential to greatly enhance the throughput of the Chemputer, while maintaining maximum flexibility in varying experimental conditions and avoiding constraints typically encountered in HTE setups. We also demonstrated how this approach can be extended for the exploration of unknown reaction spaces, combining digital discovery and optimization in a single framework. Overall, the reported system provides a universal approach for optimizing digital recipes and can accommodate any further module developments following the χDL standard. As the number of robotically generated datasets grows, we envisage that real-time telemetry data will provide an important means for data verification. Ultimately, we believe that the toolkit described herein will reduce barriers to automated process development and optimization as well as more complex, autonomous molecular discovery workflows.

## Methods

### Chemputer software

Chemputer Optimizer software package has been developed to execute closed-loop synthesis optimization using the Chemputer as automation platform. The core routine utilizes the special methods of the *DynamicStep* of the χDL framework to run the procedure iteratively, updating the user-defined parameters using the feedback from analytical instruments. Additional dynamic steps have been developed to execute chemical operations with process monitoring using various utility sensors: *DynamicAdd*—for dynamic reagent addition, *DynamicTransfer*—for crucial liquid transferring operations and *DoUntil*—for active reaction monitoring. The *AnalyticalLabware* library introduces methods to control several analytical instruments in a unified manner, including HPLC, benchtop NMR and Raman spectrometer. In addition, this library provides operations for basic processing and analysis of the acquired data. All modules are written in Python 3.9, with full source code and documentation available online as supplementary files and on GitHub. The code is compatible with existing ChemPU software stack and can be extended for use in any χDL compatible automation platform.

### Experimental setup

All experiments were executed on the ChemPU platform, equipped with 1/16" PTFE tubing. The HPLC instrument (Agilent 1260 Infinity II) was installed with an additional sample loop switching valve (Rheodyne MX Series II™) connected to the liquid handling system. The instrument was triggered after sample loading, with additional control over the experiment achieved using macro commands implemented in the AnalyticalLabware module. The benchtop NMR (Magritec Spinsolve 80 Carbon) was equipped with a flow cell and connected to the liquid handling system via threaded fitting and 1/16" PTFE tubing. The Raman spectrometer (OceanInsight QE Pro) is coupled with a 754 nm laser (OceanInsight LASER-785-LAB-ADJ-SMA) and used with a contactless probe (OceanInsight RIP-RPB-785-SMA-SMA) installed at the round bottom flask. All analytical experiment were executed via dedicated χDL steps, specifying all necessary protocol options, dilution or quenching steps. Therefore, time inconsistencies between recorded reaction time and actual reaction time are minimal systematic errors. All process sensors were connected to the Chemputer network using the SensorHub - a PCB featuring number of communication protocols and an Ethernet module for control over an IP network. The details of the setup for each experiment are given in the Supporting Information.

### Four-component Ugi reaction optimization

The original procedure was translated into χDL, amended for iterative optimization, and executed on the ChemPU platform with benchtop NMR installed. The following strategies for the parameter optimization were used: 5 experiments with random search strategy, 14 SMBO explorative experiments, 5 SMBO balanced search and 6 SMBO exploitation experiments. The reaction was analysed using $^{19}$F NMR and the optimization was set to maximize the peak of the Ugi product with respect to the 1,4-difluorobenzene as internal standard.

Three-neck 25-mL round bottom flask (reactor) equipped with reflux condenser, glass stopper, tubing connector to a liquid handling system, DrySyn© aluminium block and a magnetic stirrer bar. In the beginning of the procedure the liquid handling system was washed with methanol. Benzaldehyde (0.2 mL, 1.96 mmol) was added automatically, following by benzylamine (0.10–1.00 mL, 0.92–9.20 mmol). The reaction mixture was stirred for 0.0–30.0 min and 2-fluorobenzoic acid (2.0 M solution in methanol, 0.50–3.00 mL, 1.00–6.00 mmol) was added, following by isocyanide (0.10–1.00 mL, 1.63–16.3 mmol). The resulting mixture was stirred for 2.0–18.0 h at 25.0–60.0 °C. Thereafter the 1,4-difluorobenzene (0.2 M solution in DCM, 5.00 mL, 1.00 mmol) was added and the sample (2.5 mL) of the resulting mixture was transferred to the NMR for analysis. Upon analysis completion, the sample was transferred back to the flask and all its contents was moved to an empty flask for storage. The reactor was cleaned twice with DCM (15 mL) and used for the next iteration.

### Van Leusen oxazole synthesis optimization

The original procedure was translated into χDL, amended for iterative optimization, and executed on a Chemputer platform equipped with the HPLC and two independent reactor modules, each consisting of three-neck 25-mL round bottom flasks with reflux condenser, glass stopper, tubing connector to a liquid handling system, DrySyn© aluminium block and a magnetic stirrer bar. The optimization target was to maximize the peak area of the product with respect to naphthalene as internal standard while simultaneously minimizing impurities (see ESI Section 5.5.2 for details). The SNOBFIT algorithm was used as implemented in the Summit framework through the client-server interface.

0.25 M TosMIC in MeOH solution (4.10–6.15 mL), 0.25 M 4-formylbenzonitrile in MeOH (4.1 mL, containing 0.05 M naphthalene as an internal standard), neat DBU (0.15–0.31 mL) and methanol (5 mL) were added to the reactor. The reaction mixture was stirred for 30–180 minutes at 25.0–75.0 °C. After cooling to room temperature, a sample is withdrawn from the reactor, 40 times diluted in an empty flask, and subsequently loaded onto a 5 mL sample loop and injected into the HPLC. The remaining volume of the reaction mixture was discarded, and the platform reset by cleaning all modules with methanol and/or acetonitrile.

### Styrene sulfonate oxidation optimization

The original procedure was translated into χDL, amended for iterative optimization, and executed on a Chemputer platform equipped with Raman spectrometer monitoring a single reactor module consisting of a three-neck 25-mL round bottom flask with reflux condenser, glass stopper, tubing connector to a liquid handling system, DrySyn© aluminium block and a magnetic stirrer bar. The optimization target was to minimize the area of the peak of the double bond of the starting material at 1633 cm$^{-1}$ relative to the area of the peak at 2250 cm$^{-1}$, with respect to the peak of an internal standard (acetonitrile). The deep Bayesian optimizer Phoenics as available through the Olympus framework was chosen as the algorithm.

Five mM MnSO$_4$ in water solution (0.5–5.0 mL), 0.5 M styrene sulfonate in water solution (5.0 mL, 2.5 mmol, 1 eq.), and 0.5 M NaHCO$_3$ in water solution (0.5–5.0 mL) were automatically transferred to a reactor vessel, placed in front of a Raman probe. 30% Hydrogen

peroxide in water (1.0–5.0 mL) was added at a rate of 0.04–10.00 mL/min. The reaction mixture was stirred for 1.0–10.0 h. The process was continuously monitored via Raman and single end-point analysis was taken for calculating the optimization target value. The reaction mixture was discarded, and the platform was reset by cleaning the reactor vessel with water and a cleaning solution (volume).

## Trifluoromethylation reaction exploration

The original procedure was amended to include an alternative workup process to reduce overall experiment time. The initial exploration phase of the experiment was set to maximize the *novelty* of the product space, i.e. maximize number of novel peaks on the $^{19}$F NMR spectrum. In the second phase a set of individual optimization experiments were run with the objective set to maximize the integration area of the regions of interest on the spectrum, as identified during initial phase.

Three-neck 25-mL round bottom flask (reactor) equipped with reflux condenser, temperature probe, tubing connector to a liquid handling system, DrySyn© aluminium block (connected to a chiller) and a magnetic stirrer bar. In the beginning of the procedure the liquid handling system was washed with THF. Cyclohexenone (0.20 mL, 1.0 mmol) was added automatically, following by 4-fluoroacetophenone (0.23 mL, 1.0 mmol), butyrolactone (0.16 mL, 1.0 mmol), myrtenal (0.30 mL, 1.0 mmol) and THF (5 mL). The reaction mixture was adjusted to 22 °C and trifluouoromethyltrimethylsilane (0.9 mL, 6.1 mmol) was added, following by 0.1 M solution of TBAF in THF (0.4 mL, 0.04 mmol). The resulting mixture was stirred for 5 min at maintained temperature. Thereafter the reaction mixture was adjusted to 22 °C, 1.0 M TBAF solution in THF (3.0 mL, 3.0 mmol) was added to cleave TMS group and the reaction mixture was stirred for another 5 min. The solution of fluorobenzene (1.0 M in DCM, 2.0 mL, 2.0 mmol) was added and the sample (2.5 mL) of the resulting mixture was transferred to the NMR for analysis. Upon analysis completion, the sample was transferred back to the flask and all its contents was moved to an empty flask for storage. The reactor was cleaned twice with THF (20 mL) and used for the next iteration.

## The tosMIC and phloroglucinol reaction optimization

The original procedures have been obtained from the algorithmic reaction discovery process, amended for iterative optimization, and executed on a Chemputer platform equipped with HPLC.

The tosMIC reaction: for the parameter optimization were used: 10 experiments with random search strategy, and 22 SMBO explorative experiments. The reaction was analysed HPLC and the optimization was set to maximize the peak of the product with respect to the naphthalene as internal standard.

In all, 1.0 M TosMIC in DMSO solution (variable), 1.0 M benzylidenemalononitrile in DMSO (1 mL) and DMSO (variable) were added to the reactor. The reaction mixture was stirred for a variable amount of time at a variable temperature. After cooling to room temperature, reference was added to the mixture (1 mL 1.0 M naphthalene) and afterwards a sample (0.5 mL) is withdrawn for the reactor, transferred to an empty flask, diluted to 10.0 mL, and subsequently loaded onto a 5 μL sample loop and injected into the HPLC. The remaining volume of the reaction mixture was discarded, and the platform reset by cleaning all modules with DMSO.

The phloroglucinol reaction: for the parameter optimization were used: 10 experiments with random search strategy, and 40 SMBO explorative experiments. The reaction was analysed HPLC and the optimization was set to maximize the peak of the product with respect to the naphthalene as internal standard.

In all, 1.0 M phloroglucinol in DMSO solution (variable), 1.0 M benzylidenemalononitrile in DMSO (1 mL), 0.5 M 1,8-bis(dimethylamino)naphthalene in DMSO solution (variable) and DMSO (variable) were added to the reactor. The reaction mixture was stirred for a variable amount of time at a variable temperature. After cooling to room temperature, reference was added to the mixture (1 mL 1 M naphthalene) and afterwards a sample (0.5 mL) is withdrawn for the reactor, transferred to an empty flask, diluted to 10.0 mL, and subsequently loaded onto a 5 μL sample loop and injected into the HPLC. The remaining volume of the reaction mixture was discarded, and the platform reset by cleaning all modules with DMSO.

## Data availability

All data are available in the paper and its Supplementary Information files and Source data files. Source data are provided with this paper.

## Code availability

Raw data and code is linked and are available on Zenodo[38]: https://github.com/croningp/analyticallabware; https://github.com/croningp/chemputeroptimizer; https://github.com/croningp/summitserver/.

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

## Acknowledgements

We thank Dr Hessam Mehr, Dr Sebastian Manzano and Dr Dario Cambié for their assistance in integrating sensors and analytical hardware, Dr Ekaterina Trushina and Dr Mindaugas Siauciulis for their feedback on sensors integration, Dr Graham Keenan for general help with software development, Dr Dario Caramelli, Yibin Jiang and Daniel Kowalski for comments on the manuscript. We gratefully acknowledge financial support from the EPSRC (Grant Nos. EP/L023652/1, EP/R020914/1, EP/S030603/1, EP/R01308X/1, EP/S017046/1, and EP/S019472/1), the ERC (Project No. 670467 SMART-POM), the EC (Project No. 766975 MADONNA), and DARPA (Project Nos. W911NF-18-2-0036, W911NF-17-1-0316, and HR001119S0003). AJSH acknowledges a scholarship from the Studienstiftung des deutschen Volkes.

## Author contributions

LC conceived the concept, the architecture, and the programming approach. AIL developed the ChemputationOptimizer and dynamic execution software with help from AJSH. AIL, SHMM, DC, AJSH, DA, SL and SOS configured the platform, executed the procedures, and characterized the products for the following experiments: AIL—Ugi reaction optimization and trifluoromethylation exploration; AJSH—dynamic execution experiments and oxazole reaction optimization; AJSH and DA—epoxidation reaction optimization, SL and SOS—tosMIC and phloroglucinol reactions. AK developed computer vision system. MC contributed to the initial development of the dynamic execution software and dashboard for sensor data visualization. MC, LW, AJSH and AIL developed the parallelization routine. The manuscript was written by LC together with AIL and AJSH with input from all the authors.

## Competing interests

A patent based on this work has been filed by the University of Glasgow, United Kingdom (GB) Patent Application No: 2315721.7. There are no other competing interests.
