## [Peer Review File · Nature Communications]

REVIEWER COMMENTS

Reviewer #1 (Remarks to the Author):

In this manuscript, Cronin et al. reported a self-optimizing chemical synthesis system. Compared to previously demonstrated systems, this system features the integration of several low-cost sensors to achieve online monitoring of various parameters during the automated reaction. In addition, self-optimization algorithms were implemented to drive reaction optimizations and exploration. The effort to integrate sensors and optimization algorithms certainly advances the field of chemistry automation, making the system to be more capable and human-like. Overall, it is a high-quality system development work, but the following points should be addressed before being considered to publish in Nature Communication:

(1) Integrating pH, conductivity, colorimetry, and temperature sensors in chemical synthesis system is widely exercised by chemists or chemical engineers to monitor the status of the reaction. It is appreciated that authors used these sensors to facilitate the operation of system, including syringe failure detection, reagent addition control, and reaction monitoring. But it seems these sensors did not help with automated reaction optimization discussed in the later part of the manuscript. This leads to the lack of synergy between implementation of sensors and automated reaction optimization. One would expect reaction optimization can be accelerated by using these sensors. For example, chemists sometimes can use color to tell whether a catalyst is dead, and thus infer the reaction outcome without time-consuming chromatograph analysis.

Thus, it would be nice to see an example to demonstrate the synergy between installed sensors and optimization algorithms.

(2) The closed-loop optimizations demonstrated in this work involve the reaction time. However, it seems that the quenching steps are missing for the demonstrated optimization examples in the experimental procedure, which may lead to inconsistency between the actual reaction time and the recorded reaction time.

(3) In the phloroglucinol, benzylidenemalononitrile and 1,8-bis(dimethylamino)naphthalene reaction, second campaign with expanded search space was executed after first campaign, which led to a further increase in yield. Search space modification during the optimization campaign is also discussed in "Torres J A G, Lau S H, Anchuri P, et al. A Multi-Objective Active Learning Platform and Web App for Reaction Optimization. *Journal of the American Chemical Society*, 2022, 144(43): 19999-20007." They used the Expected Improvement (EI) indicator for deciding when to expand the search space. It should be added in the reference.

(4) In the Figure 6, the subgraph b and d are not very intuitive. Other illustration approaches should be considered to make this figure more informative.

(5) Since it is a long manuscript containing multiple topics of the demonstrated system, it is recommended to organize the manuscript with subtitles for clearer content hierarchy.

Reviewer #2 (Remarks to the Author):

The authors present an evolution of their previously published platform with improved functionalities achieved thanks to the use of a broad range of sensors. The study presents a number of applications including self-optimisation and discovery algorithms. The work will be of interest to the scientific community working on the field, in particular the availability of the open-source code, macros, etc. However, several shortcomings should be addressed prior to publication:

1. The platform builds on extensive work of the group on the concept and applications of the "chemputer". In the work presented is not completely clear what are the novel and unique aspects introduced. This should be clearly stated.
2. Some figures are not very informative (e.g. Fig. 6b and 6d), or have minor defects (e.g. Figure 3, top right box shows "aglorithmic discovery" on what appears to be an aldol condensation reaction, which is well known). These aspects can be easily corrected and will help to convince the traditional organic chemistry community of the benefits of digitalisation.
3. The choice of reactions, algorithms and conditions seems a bit random at the moment. This is to some extent due to the broad range of examples showed by the authors. A better explanation of the rationale employed for the selection of the different procedures selected and implemented would benefit the manuscript.
4. The authors claim that at points the algorithm selected non-intuitive conditions, like a temperature at which DMSO should be solid. How did the platform handle such conditions?

Reviewer #3 (Remarks to the Author):

This is a very exciting paper showing further development of the ChemPU system developed by the Cronin group. The incorporation of new sensing technologies and a series of different reaction

classes is far beyond an incremental improvement and I fully recommend the article for publication. The article nicely expands many of the approaches used in continuous reactors to an automated batch system and includes several new advancements in process sensing, reaction classes studied and error handling.

Major edits:

- i) As the authors have developed a vision based monitoring system (and are reporting it here). Can the authors provide a critical analysis of the occurrence of errors and hardware failures when using the system for the work presented in the paper.
- ii) The authors are often not very specific with the reporting of reaction yields. Can they edit the document throughout and be more specific e.g. on p10 "in 30 experiments... to achieve a 38% improvement" is this an absolute figure or relative and what was the final yield? I know the data is in the ESI but please add in main text. (similar on p12 with snobfit example "achieving a 10% improvement. And p13 with the SMBO work)
- iii) On p12 the authors state that 40 iterations were needed to find a robust optimum that leads to full conversion. Can the authors provide some commentary on whether this was optimal for product yield too? (i.e. not just SM conversion).
- iv) The authors provide a set of dropbox files for the review. Please confirm these will be hosted on a persistent archive/journal website?
- v) The stand alone python package "AnalyticalLabware" is this hosted with the paper/github (I see it is part of the reviewers files"? Perhaps this should have also have a online reference in the text pointing to both online file repository and current version uploaded with paper if this is the intention.
- vi) The parallel reactors used with the snobfit algorithm are very interesting. Can the authors comment more in the main text about acceleration of optimisation? Was it twice the rate or more/less?

Typos:

P8 "and addition left distinct signals" doesn't make sense, please reword

P6 replace "next setup" with "next set of input conditions" or similar

Figure 5 – I like most of this plot, but Figure 5a plot with the parameter space reduced to two dimensions is not very useful in my opinion, can the authors swap the color for the target function and use the symbol shape to denote the type of point instead similar to figure 5d?

Figure 6 -b and d are both parallel coordinate plots and this should be added in description in figure caption. This figure caption language is perhaps too tailored for expert readers and the schemes, optimisation progress and parallel coordinate plots could be described more clearly for less familiar

readers. Could the second campaign also be differentiated in the plots (perhaps be a dotted line) in the progress vs experiment number plots.

Point by point reply to the reviewer comments. Reviewer comments in italics. Our replies in normal type.

Reviewer #1 (Remarks to the Author):

In this manuscript, Cronin et al. reported a self-optimizing chemical synthesis system. Compared to previously demonstrated systems, this system features the integration of several low-cost sensors to achieve online monitoring of various parameters during the automated reaction. In addition, self-optimization algorithms were implemented to drive reaction optimizations and exploration. The effort to integrate sensors and optimization algorithms certainly advances the field of chemistry automation, making the system to be more capable and human-like. Overall, it is a high-quality system development work, but the following points should be addressed before being considered to publish in Nature Communication:

(1) Integrating pH, conductivity, colorimetry, and temperature sensors in chemical synthesis system is widely exercised by chemists or chemical engineers to monitor the status of the reaction. It is appreciated that authors used these sensors to facilitate the operation of system, including syringe failure detection, reagent addition control, and reaction monitoring. But it seems these sensors did not help with automated reaction optimization discussed in the later part of the manuscript. This leads to the lack of synergy between implementation of sensors and automated reaction optimization. One would expect reaction optimization can be accelerated by using these sensors. For example, chemists sometimes can use color to tell whether a catalyst is dead, and thus infer the reaction outcome without time-consuming chromatograph analysis. Thus, it would be nice to see an example to demonstrate the synergy between installed sensors and optimization algorithms.

We thank the reviewer for this suggestion. This is indeed an interesting avenue for future work that may be communicated separately. Preliminary work indicated that low-cost sensors can be useful in bespoke and well understood systems (such as the RGB sensor example in the manuscript) but generally are not as universally applicable as chromatographic analysis that allow for deeper insight into the chemistry. Furthermore, even for a “failed” reaction with a dead catalyst, it would still be beneficial to measure the reaction outcome to assess if / how much product formed rather than skipping the analysis. The examples in the paper are representative of >90% of use cases that we expect practitioners to encounter.

(2) The closed-loop optimizations demonstrated in this work involve the reaction time. However, it seems that the quenching steps are missing for the demonstrated optimization examples in the experimental procedure, which may lead to inconsistency between the actual reaction time and the recorded reaction time.

Reactions were typically quenched either by rapid dilution or addition of reagents during work-up steps during the analysis. These are part of the high-level Analyze XDL steps. For Raman, no quenching is needed as the measurement is instant. Therefore, time inconsistencies are minimal systematic errors. Executing the optimized, versioned XDL codes on our platform leads to reproducible optimal results.

(3) In the phloroglucinol, benzylidenemalononitrile and 1,8-bis(dimethylamino)naphthalene reaction, second campaign with expanded search space was executed after first campaign, which led to a further increase in yield. Search space modification during the optimization campaign is also discussed in “Torres J A G, Lau S H, Anchuri P, et al. A Multi-Objective Active Learning Platform and Web App for Reaction Optimization. Journal of the American Chemical Society, 2022, 144(43): 19999-20007 .” They used the Expected Improvement (EI) indicator for deciding when to expand the search space. It should be added in the reference.

We thank the reviewer for this comment and added the reference to our discussion.

(4) In the Figure 6, the subgraph b and d are not very intuitive. Other illustration approaches should be considered to make this figure more informative.

We think the figure gives the information needed but the figure caption was very confusing. We therefore have rewritten the caption to explain the figure properly and to help the reader have the intuition to understand the figure.

(5) Since it is a long manuscript containing multiple topics of the demonstrated system, it is recommended to organize the manuscript with subtitles for clearer content hierarchy.

We thank the reviewer for this comment and added subtitles to help guide the reader.

Reviewer #2 (Remarks to the Author):

The authors present an evolution of their previously published platform with improved functionalities achieved thanks to the use of a broad range of sensors. The study presents a number of applications including self-optimisation and discovery algorithms. The work will be of interest to the scientific community working on the field, in particular the availability of the open-source code, macros, etc. However, several shortcomings should be addressed prior to publication:

1. The platform builds on extensive work of the group on the concept and applications of the "chemputer". In the work presented is not completely clear what are the novel and unique aspects introduced. This should be clearly stated.

We thank the reviewer for the comment. The introduction and conclusion paragraphs were modified to clarify the novelty of the work, namely the Dynamic XDL as well as the automated tandem of discovery-optimization framework.

2. Some figures are not very informative (e.g. Fig. 6b and 6d), or have minor defects (e.g. Figure 3, top right box shows "aglorithmic discovery" on what appears to be an aldol condensation reaction, which is well known). These aspects can be easily corrected and will help to convince the traditional organic chemistry community of the benefits of digitalisation.

The descriptions for Figures 6b and 6d have been clarified. The top right panel on Figure 3 has been exchanged to represent the actual discovery optimization process.

3. The choice of reactions, algorithms and conditions seems a bit random at the moment. This is to some extent due to the broad range of examples showed by the authors. A better explanation of the rationale employed for the selection of the different procedures selected and implemented would benefit the manuscript.

The goal was to cover a broad change of chemistry incl. heterocycle synthesis, metal-catalysed reactions, functional group interconversions, and multicomponent reactions. We have clarified this intention in the text.

4. The authors claim that at points the algorithm selected non-intuitive conditions, like a temperature at which DMSO should be solid. How did the platform handle such conditions?

The algorithm will gain information from both successful and failed reactions to find optimal conditions within the given search space. This can include non-intuitive conditions; however human researchers have the ability to encode expert knowledge by selecting appropriate bounds for the search space. The platform itself had no issues with handling these conditions. Thanks to continuous stirring, DMSO was kept in a slurry-like state, allowing for reagent mixing and reaction continuation.

Reviewer #3 (Remarks to the Author):

This is a very exciting paper showing further development of the ChemPU system developed by the Cronin group. The incorporation of new sensing technologies and a series of different reaction classes is far beyond an incremental improvement and I fully recommend the article for publication. The article nicely expands many of the approaches used in continuous reactors to an automated batch system and includes several new advancements in process sensing, reaction classes studied and error handling.

Major edits:

As the authors have developed a vision based monitoring system (and are reporting it here). Can the authors provide a critical analysis of the occurrence of errors and hardware failures when using the system for the work presented in the paper.

Hardware failures are generally very rare events. The computer vision system was built for one of the most frequently occurring errors, namely syringe pump failure. In the course of the work, no such failures were observed when the platform was using the vision-based monitoring system. This could be due to the relatively mild reaction conditions employed in most examples (compared to harsher conditions used in our other work published in *Science*, **2022**, 377, 172-180 which motivated this development). We have plans to expand on the vision system as we continue to collect data and communicate these developments separately. We added a failure analysis to the supporting information (Section 5.3.1).

The authors are often not very specific with the reporting of reaction yields. Can they edit the document throughout and be more specific e.g. on p10 “in 30 experiments... to achieve a 38% improvement” is this an absolute figure or relative and what was the final yield? I know the data is in the ESI but please add in main text. (similar on p12 with snobfit example “achieving a 10% improvement. And p13 with the SMBO work)

The information was added in the main text for clarity.

On p12 the authors state that 40 iterations were needed to find a robust optimum that leads to full conversion. Can the authors provide some commentary on whether this was optimal for product yield too? (i.e. not just SM conversion).

In this specific example, the objective was to optimize for SM conversion and product yield was not measured during the optimization campaign. However, using the champion recipe we obtained the desired product in excellent yield. This path has been chosen due to the following facts: (i) the substrate was easier to observe than the product and (ii) increased reaction times led to the formation of 1, 2-diol – a side product we wanted to avoid.

The authors provide a set of dropbox files for the review. Please confirm these will be hosted on a persistent archive/journal website?

Reply: The dropbox link in the manuscript has been substituted with three links to GitHub repositories that will be public upon the paper's acceptance.

<https://github.com/croningp/analyticallabware>
<https://github.com/croningp/chemputeroptimizer>
<https://github.com/croningp/summitserver/>

The stand alone python package “AnalyticalLabware” is this hosted with the paper/github (I see it is part of the reviewers files)? Perhaps this should have also have a online reference in the text pointing to both online file repository and current version uploaded with paper if this is the intention.

The “AnalyticalLabware” package is one of the repositories mentioned in the question above and will be provided via GitHub.

The parallel reactors used with the snobfit algorithm are very interesting. Can the authors comment more in the main text about acceleration of optimisation? Was it twice the rate or more/less?

The expected acceleration is less than 2x due to multiple processes relying on the same hardware resources. For example, the 2-fold parallel system had an estimated 1.6X acceleration and a 4-fold parallel system exhibited a 2.7x acceleration in simulations. It is important to note that these factors can vary based on the experiments selected by the algorithm (e.g. different combinations of short and long reactions times allowing for different degrees of concurrent execution). In practice, the acceleration depends on the accuracy of the

expected durations for the different XDL steps that are used by the scheduling routine. In our hands, this led to factors of $< 1.6X$, however this is expected to improve as we collect more data on actual durations. We added a comment in the supporting information.

Typos:

P8 "and addition left distinct signals" doesn't make sense, please reword

P6 replace "next setup" with "next set of input conditions" or similar

Figure 5 – I like most of this plot, but Figure 5a plot with the parameter space reduced to two dimensions is not very useful in my opinion, can the authors swap the color for the target function and use the symbol shape to denote the type of point instead similar to figure 5d?

Figure 6 -b and d are both parallel coordinate plots and this should be added in description in figure caption. This figure caption language is perhaps too tailored for expert readers and the schemes, optimisation progress and parallel coordinate plots could be described more clearly for less familiar readers. Could the second campaign also be differentiated in the plots (perhaps be a dotted line) in the progress vs experiment number plots.

We have fixed all the typos. Figure 5 has been altered according to the reviewer's suggestion. The caption for Figures 6b and d has been improved for readability and explanation. As suggested, the second campaign has been denoted using a dashed line.

REVIEWERS' COMMENTS

Reviewer #1 (Remarks to the Author):

Overall, I think the authors addressed most of the comments raised by the reviewers.

Reviewer #2 (Remarks to the Author):

The authors have satisfactorily addressed the reviewers' comments. I recommend to publish the manuscript.

Reviewer #3 (Remarks to the Author):

The authors have addressed the comments well and I support publication of the article (following the revisions outlined below).

Comment 2 by reviewer 1 (regarding the quench and analysis times) should also be addressed in the manuscript text so that other readers are also informed of this. I would suggest information about the quench processes is added to the SI too.

I would also like to check the github repositories are suitable before acceptance as I believe these should be of high quality.

I would suggest in the future that the text edits are copied into the response to reviewers as this massively reduces the burden of reviewing.